# Factors Associated with Mortality among Elderly People in the COVID-19 Pandemic (SARS-CoV-2): A Systematic Review and Meta-Analysis

**DOI:** 10.3390/ijerph18158008

**Published:** 2021-07-29

**Authors:** Vicente Paulo Alves, Francine Golghetto Casemiro, Bruno Gedeon de Araujo, Marcos André de Souza Lima, Rayssa Silva de Oliveira, Fernanda Tamires de Souza Fernandes, Ana Vitória Campos Gomes, Dario Gregori

**Affiliations:** 1Stricto Sensu Graduate Program in Gerontology/Medicine, University Catholic of de Brasília, Taguatinga 71966-700, Brazil; brunogedeon@gmail.com (B.G.d.A.); marcosandreiteb@gmail.com (M.A.d.S.L.); rayssaolimed@gmail.com (R.S.d.O.); fernandatamires.sf@gmail.com (F.T.d.S.F.); anavitoriag@hotmail.com (A.V.C.G.); 2Unit of Biostatistics, Epidemiology and Public Health, Padova University, 35122 Padova, Italy; dario.gregori@unipd.it; 3Ribeirão Preto School of Nursing, University of São Paulo, São Paulo 14040-902, Brazil; francine_gc@hotmail.com

**Keywords:** SARS-CoV-2, COVID-19, non-communicable chronic diseases (NCCDs), clinical features, institutionalized or hospitalized elderly, meta-analysis

## Abstract

The objective of this meta-analysis was to evaluate the factors associated with the mortality of elderly Italians diagnosed with coronavirus who resided in institutions or who were hospitalized because of the disease. Methods: A systematic review following the recommendations of The Joanna Briggs Institute (JBI) was carried out, utilizing the PEO strategy, i.e., Population, Exposure and Outcome. In this case, the population was the elderly aged over 65 years old, the exposure referred to the SARS-CoV-2 pandemic and the outcome was mortality. The National Center for Biotechnology Information (NCBI/PubMed), Latin American and Caribbean Literature in Health Sciences (LILACS), Excerpta Medica Database (EMBASE) and Cumulative Index to Nursing and Allied Health Literature (CINAHL) databases were used until 31 July 2020. Results: Five Italian studies were included in this meta-analysis, with the number of elderly people included varying between 18 and 1591 patients. The main morbidities presented by the elderly in the studies were dementia, diabetes, chronic kidney disease and hypertension. Conclusions: The factors associated with the mortality of elderly Italian people diagnosed with SARS-CoV-2 who lived in institutions or who were hospitalized because of the disease were evaluated. It was found that dementia, diabetes, chronic kidney disease and hypertension were the main diagnosed diseases for mortality in elderly people with COVID-19.

## 1. Introduction

One of the greatest achievements of humanity has been longevity, which—although there are still differences between countries influenced by the socioeconomic context of each—in general, is led by progress in the population’s health markers. Achieving old age, which was once the privilege of a few, has now become one of the world’s main goals and challenges [1].

In this way, aging has become a global phenomenon and a success story for public health policies and socioeconomic development. However, there are new challenges for society that this presents. Our society needs to adapt to this new scenario, to maximize the functional capacity and health of the elderly and to promote their social inclusion and safe participation [1]. In view of this, there are social consequences of the aging population and new public health issues arising that affect European countries, such as Italy, in particular [2]. In Italy, the profile of the elderly population is of a group with a high prevalence of non-communicable chronic diseases (NCCDs) and associated comorbidities [1]. In Italy, aging is a common and growing phenomenon. Italy is considered the country with the second largest number of elderly people [2], along with a mortality rate that has decreased by more than 50% in the last 30 years, mainly due to the reduction in cardiovascular diseases [3].

The COVID-19 pandemic (SARS-CoV-2) has caused considerable mortality in populations considered at risk, such as the elderly population, especially those who are institutionalized, a scenario in which social isolation is difficult in a situation such as a pandemic. The vulnerability of this population is linked with the physiological aspects of aging, which impact the effectiveness of the immune system, triggering morbidity and mortality from infectious diseases [4].

Thus, it is necessary to investigate the main factors that make institutionalized elderly people more vulnerable to death. Fragility is a condition that worsens with advancing age and with COVID-19 infection, especially for the hospitalized elderly, who tend to develop a more accentuated presentation of the classic symptoms of the disease [5].

Since the onset of the COVID-19 pandemic, several studies have begun to be carried out in different contexts, generating scientific evidence and statistical data that point in certain directions. It has become a challenge for all related researchers to contribute to the advancement of knowledge regarding the reach of COVID-19 and the factors associated with mortality. In this specific case, it was proposed that we carry out a systematic review of studies published in Italy between March and July 2020 to identify such factors. The decision to select texts that were published from Italian research was made as local transmission first took place in Italy before the spread of COVID-19 went on to impact other European regions [6].

In Italy, the contagion’s outbreak started on 20 February 2020, with the number of confirmed cases increasing by 428% in the following 30 days. Residential facilities for the elderly were the hardest hit, according to data released by the Istituto Superiore di Sanità (ISS) [7]. The elderly who died in these residential establishments due to COVID-19, who underwent the reverse transcription-polymerase chain reaction (RT-PCR) tests to confirm their infection, represent around 7.4% of all deaths from the period. When adding all those who died with flu symptoms (without an objective assessment) to these data, the number of deaths represents 41.2%. A survey was carried out by the ISS in July 2020, which sent a questionnaire to 3417 establishments, of which 1356 responded, accounting for a total of 97,521 elderly living residents [8].

The objective of this study was to synthesize the factors associated with the mortality of elderly Italian people diagnosed with coronavirus who lived in institutions or who were hospitalized because of the disease.

## 2. Materials and Methods

### 2.1. Search Strategy and Selection Criteria

The systematic review format was chosen based on the recommendations of The Joanna Briggs Institute (JBI), following the nine steps for its development: (1) construction of the preliminary research protocol; (2) formulation of the review question; (3) definition of inclusion and exclusion criteria; (4) search strategy; (5) selection of studies for inclusion; (6) data extraction; (7) synthesis of the data; (8) narrative summary; (9) references [9].

This study used public, free-to-access articles located in databases of scientific literature. Primary studies on the mortality of elderly Italians with a diagnosis of coronavirus were selected, with publications in the English, Italian and Spanish languages included, which carried out quantitative and qualitative analyses. To formulate the research question, the PEO strategy was used, i.e., Population, Exposure and Outcome [10]. It was determined that the “Population” (P) would be the elderly aged over 65 years, the “Exposure” (E) would be the SARS-CoV-2 pandemic and the “Outcome” (O) would be mortality. Thus, the guiding question of this study was: “What are the factors related to the mortality of Italian elderly people diagnosed with the COVID-19 (SARS-CoV-2) disease?”

The inclusion criteria for the selection of articles were:Primary studies on the mortality of elderly people diagnosed with coronavirus;Studies in English, Spanish or Italian.

Once the inclusion criteria were established, these were set as the exclusion criteria:Studies that were not of Italian elderly people;Studies on the elderly who were not institutionalized or hospitalized;Studies that did not answer the guiding question of the systematic review.

### 2.2. Data Extraction and Analysis

The search for publications was carried out in July 2020 in the following databases: The National Center for Biotechnology Information (NCBI/PubMed), Latin American and Caribbean Literature in Health Sciences (LILACS), Excerpta Medica Database (EMBASE) and Cumulative Index to Nursing and Allied Health Literature (CINAHL). The search strategy combined controlled and uncontrolled descriptors, according to the indication offered in each database. To search for articles on PubMed, controlled descriptors from Medical Subject Headings (MeSH) were used; Heading-MH was consulted for the CINHAL database; Embase subject headings (EMTREE) were used to search EMBASE; Health Sciences Descriptors (DeCS) was used to search LILACS. For these searches, “aged”, “coronavirus infections” and “mortality” were used. The Boolean operator “AND” was used in all combinations as follows: “aged AND mortality AND coronavirus infections”. There was no time limit for publication. For the selection of articles, the Rayyan application, developed by the QCRI (Qatar Computing Research Institute), was used, which helps in systematic reviews by facilitating the selection process for reading articles. That process took place in three stages: in the first stage, the databases were searched; secondly, the title and abstract were read: two researchers performed a bibliographic search and independently extracted data from the included studies, where disagreements were resolved by a third investigator or by consensus, with the aim being to identify studies for the third stage; thirdly, the articles were read in full, with the aim of selecting those that were in agreement with the inclusion criteria [11].

While developing the search and selection of articles, from searching the databases to selecting studies by reading titles and abstracts or the full text, the PRISMA protocol was used [12] (Figure 1) to guarantee the rigor of the systematic review [11].

From the findings, the results were organized by performing a descriptive synthesis of the data, as shown in Table 1.

The meta-analysis was conducted using Stata software, version 16.0. Initially, the mortality rate was estimated using the number of deaths as the numerator and the total number of analyzed samples as the denominator, multiplied by the constant 100%. A grouped meta-analysis of the mortality rate was performed using random effects models [13]. The heterogeneity of the studies was assessed using the I-square (I^2^) statistic [14].

Next, the factors associated with mortality were analyzed, with the outcome being death. Thus, two groups were compared (non-survivors versus survivors). The following quantitative variables were considered as predictors: age and the Charlson Index. For studies that presented data such as the median and interquartile range (IIQ) [15], these were transformed into the mean and standard deviation (SD) [16]. The following qualitative variables were considered predictors: male gender, chronic diseases, cancer, diabetes, cardiovascular diseases, chronic obstructive pulmonary disease (COPD), immunodeficiency, chronic kidney disease (CKD), metabolic disease, obesity, hypertension, familial hypercholesterolemia (FH), dementia and smoking. Variables related to the use of drugs and therapies were excluded from the risk factor analyses since this review does not address clinical trials.

The effect size was reported as the standardized mean difference (SDM) for quantitative variables or the relative risk (RR) for qualitative variables. All of these measures were followed up with a 95% confidence interval [17]. The heterogeneity between studies was assessed using the I-square (I^2^) statistic [14]. Fixed or random effects models were used depending on the heterogeneity. Variables with a *p*-value < 0.05 were considered statistically significant.

The protocol for this article was published in the International Prospective Register of Systematic Reviews, PROSPERO, in August 2020, under the register: CRD42020201790.

## 3. Results

The number of elderly people included in each study varied between 18 [18] and 1591 [19] patients. The objectives of the publications were similar, i.e., conducting a descriptive analysis of the elderly and the factors associated with coronavirus.

The main morbidities presented by the elderly in the studies were: dementia [20], diabetes [19,21], chronic kidney disease [19] and hypertension [21], showing that NCCDs had a key role to play in these cases.

Figure 2 shows the meta-analysis of the mortality rate found. A mortality rate of 27.7% was observed (95% CI, 15.7–41.57%), with high heterogeneity between studies (I^2^, 97.71%; *p* < 0.001).

The meta-analysis was conducted for each predictor variable, stratified into quantitative and qualitative variables.

Table 2 shows the descriptive analysis of the quantitative variables according to the survivors and non-survivors, and Table 3 shows the effect size, in SDM and 95% CI, of the variables affecting mortality.

The Analysis of quantitative variables showed that mortality increased with increasing age (SMD, 3.10; 95% CI, 2.79; 3.40) and Charlson Index scores (SMD, 1.74; 95% CI, 1.56; 1.92) (Table 2).

Table 4 shows the descriptive analysis of qualitative variables according to the survivors and non-survivors, and Table 5 shows the effect size, in RR and 95% CI, of the variables affecting mortality.

The analysis of quantitative variables showed that the risk of mortality was higher in individuals with diabetes (RR, 1.90; 95% CI, 1.53; 2.37), COPD (RR, 2.19; 95% CI, 1.54; 3.10), chronic kidney disease (RR, 3.96; 95% CI, 2.65; 5.91), hypertension (RR, 1.37; 95% CI, 1.24; 1.51), FH (RR, 3.27; 95% CI, 2.49; 4.29) or dementia (RR, 3.67; 95% CI, 2.43; 5.55) (Table 4).

## 4. Discussion

This study aimed to synthesize the factors associated with the mortality of elderly Italians diagnosed with coronavirus who were institutionalized or hospitalized. The data show that diabetes, chronic obstructive pulmonary disease, hypertension and dementia were morbidities that considerably increased the risk of death in the elderly. This association is presumed to be related to the high prevalence of these diseases in the elderly population [22].

The Instituto Nazionale di Statistica (ISTAT) of Italy, in its 4 May 2020 report, states that the impact of COVID-19 is greater in people with extremely compromised health conditions, causing the mortality of these people to occur in a shorter time. The document also reports that, in some cases, COVID-19 may not be the leading cause of death, but a contributing factor to overall mortality [7]. There are a series of phenomena and dynamics that affect the current state of health of Italians, such as the aging of the population, the increase in risk factors (including NCCDs), “the phenomenon of vaccination hesitation, the threat of antimicrobial resistance, the difficulties of access to innovation, the shortage of doctors, the lack of regional homogeneity and the delay in digitizing the health system that affect the system as a whole” [23].

Italy has the lowest prevalence rate, by age, for chronic obstructive pulmonary disease and cardiovascular diseases [23]. This may be fortunate as these were the diseases that increased mortality among the elderly in the articles analyzed. This link is supported by a review that described the association between cardiovascular diseases and an increased risk of complications from COVID-19 [24].

On the other hand, the country has the highest prevalence rate, by age, for dementia. As aging progresses, the risk of this diagnosis increases. It is a progressive neurodegenerative syndrome characterized by a cognitive decline that limits social functions and activities of daily living [25]. In addition to having an important impact on the quality of life of these people, dementia was also shown to be a risk factor for mortality in elderly people with COVID-19.

## 5. Conclusions

We conclude with the belief that the objective proposed for this study had been achieved, i.e., to synthesize the factors associated with the mortality of elderly Italians diagnosed with coronavirus who lived in institutions or were hospitalized because of the disease. Looking ahead, it is expected that public policies will be developed for the new reality of humanity profoundly marked by the pandemic.

NCCDs, when associated with SARS-CoV-2, are factors in the deaths of the elderly. Data relating to NCCDs are, therefore, fundamental for the elaboration of public policies and health promotion practices and the prevention of chronic diseases throughout aging. In addition, prevention strategies against coronavirus for the elderly population with NCCDs, such as chronic obstructive pulmonary disease or dementia, must be planned with a clear and precise target to prevent so many deaths from occurring among the elderly.

Certainly, we should not create more institutions that house elderly people without taking into account the greater risks that life in a large community has for the coexistence with and contagion of these diseases. It will be necessary to think creatively about new living spaces and new ways of handling work as health professionals and operators in these establishments.

The vaccination priorities for the institutionalized elderly, as established by all governments, were touched by the social movement that reverberated around the world when several army trucks transported burial coffins in the Italian city of Bergamo in March 2020. The mortality of the elderly who lived in socio-sanitary care institutions or who were taken into hospitals showed the true danger of NCCDs meeting SARS-CoV-2.

With vaccination slowly arriving in each country, as the pharmaceutical industry works to deliver enough doses and countries strive to implement efficient logistics for the distribution and application of the drug, it is hoped that all of this will pass, and that this time of great pain and suffering for many families will facilitate our learning and the growth of authorities and new public policies aimed at protecting the elderly.

The most important limitation of this research is the small number of articles found in Italy, which prevented further analysis. In future studies, factors related to chronic diseases should be considered since these aspects impact the mortality of elderly people with COVID-19.

## Figures and Tables

**Figure 1 ijerph-18-08008-f001:**
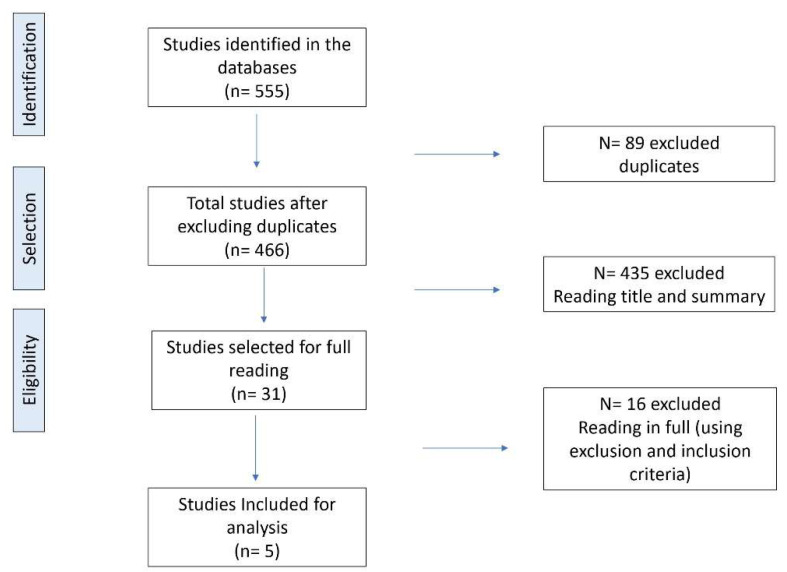
Flow diagram of the number of studies selected and included in the meta-analysis.

**Figure 2 ijerph-18-08008-f002:**
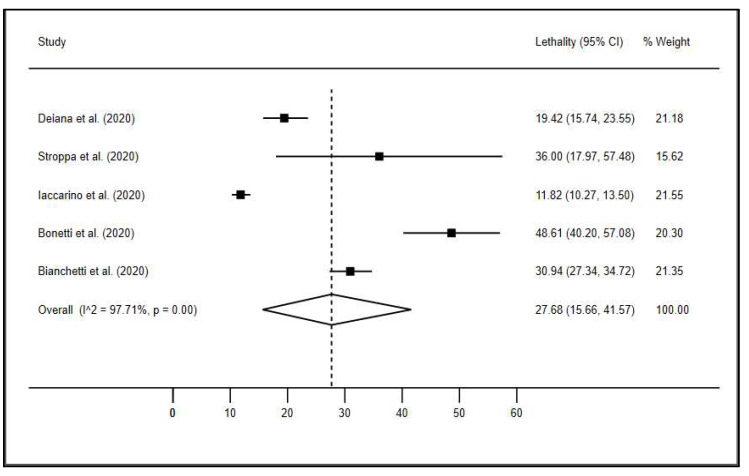
Mortality rate in the elderly obtained in the meta-analysis.

**Table 1 ijerph-18-08008-t001:** Descriptive synthesis of the data.

Author/Year	Journal	Aim	Elderly Sample	Sample Location
Bianchetti et al. (2020)	*Journal of Nutrition, Health and Aging*	To evaluate the prevalence, clinical characteristics and outcomes of dementia in individuals hospitalized for infection with COVID-19.	627	Hospitals and nursing homes in the province of Brescia, Northern Italy.
Stroppa et al. (2020)	*Future Oncology*	To describe the cases of 25 cancer patients who were infected with COVID-19.	18	Piacenza’s general hospital, Northern Italy.
Deiana et al. (2020)	*International Journal of Environmental Research and Public Health*	To describe the clinical characteristics of patients who died after a positive test for the SARS-CoV-2 infection and evaluate the influence of health conditions associated with death as the outcome.	573	Sardinia, Italy
Bonetti et al. (2020)	*Clinical Chemistry and Laboratory Medicine (CCLM)*	To describe laboratory findings in a group of Italian patients with COVID-19 in the Valcamonica area and correlate the abnormalities with the severity of the disease.	518	Emergency Department of the Valcamonica Hospital (Esine, Brescia, Lombardia, Italy).
Iaccarino et al. (2020)	*Hypertension*	To check if renin and angiotensin, the system inhibitors, are related to serious outcomes of COVID-19 infection.	1591	Multicenter study.

**Table 2 ijerph-18-08008-t002:** Descriptive analysis of quantitative variables, according to groups of survivors and non-survivors.

Variables	Non-Survivors			Survivors		
	N	Mean	SD	N	Mean	SD
Age (years)						
Iacarinno et al. (2020)	188	79.6	0.8	1304	64.7	0.4
Stroppa et al. (2020)	9	74.44	7.21	16	68.38	10.16
Bonetti et al. (2020)	70	75.4	14.99	74	62.63	14.97
Charlson Index						
Iacarinno et al. (2020)	188	4.37	0.14	1403	2.63	0.05

N, sample size in each group; SD, standard deviation.

**Table 3 ijerph-18-08008-t003:** Meta-analysis of factors (quantitative variables) associated with mortality.

Variables	SMD (95% CI)	I^2^	Z	*p*-Value
Age (years)	3.10 (2.79; 3.40)	99.9%	19.76	<0.001
Charlson Index	1.74 (1.56; 1.92)	-	19.33	<0.001

SMD, standardized mean difference; Z, Z statistic of the meta-analysis; I^2^, I-square; 95% CI, 95% confidence interval.

**Table 4 ijerph-18-08008-t004:** Descriptive analysis of qualitative variables according to groups of survivors and non-survivors.

Variables	Non-Survivors			Survivors		
	N	n	%	N	N	%
Male						
Bonetti et al. (2020)	70	45	64.3	74	51	68.9
Iacarinno et al. (2020)	188	125	66.5	1403	891	63.5
Stroppa et al. (2020)	9	5	55.6	16	15	94.8
Deiana et al. (2020)	81	40	50.6	336	89	26.6
Chronic diseases						
Bonetti et al. (2020)	70	49	70.0	74	43	57.3
Cancer						
Bonetti et al. (2020)	70	9	12.9	74	6	8.0
Diabetes						
Bonetti et al. (2020)	70	21	30.0	74	16	21.3
Iacarinno et al. (2020)	188	61	32.4	1403	208	14.8
Stroppa et al. (2020)	9	2	22.2	16	6	37.5
Cardiovascular diseases/coronary artery disease						
Bonetti et al. (2020)	70	38	54.3	74	33	44.0
Iacarinno et al. (2020)	188	56	29.8	1403	160	11.4
COPD ^1^						
Bonetti et al. (2020)	70	14	20.5	74	6	8.0
Iacarinno et al. (2020)	188	28	14.9	1403	94	6.7
Stroppa et al. (2020)	9	3	33.3	16	4	25.0
Immunodeficiencies						
Bonetti et al. (2020)	70	2	2.8	74	0	0.0
Chronic kidney disease						
Bonetti et al. (2020)	70	9	12.9	74	3	4.0
Icarinno et al. (2020)	188	31	16.5	1403	56	4.0
Metabolic disease						
Bonetti et al. (2020)	70	10	14.3	74	7	9.3
Obesity						
Bonetti et al. (2020)	70	12	17.1	74	5	6.8
Iacarinno et al. (2020)	188	12	6.4	1403	90	6.4
Hypertension						
Iacarinno et al. (2020)	188	138	72.9	1403	737	52.5
Stroppa et al. (2020)	9	5	55.6	16	11	68.8
FH ^2^						
Iacarinno et al. (2020)	188	57	30.3	1403	130	9.3
Dementia						
Bianchetti et al. (2020)	194	51	26.3	433	31	7.2
Smoking						
Stroppa et al. (2020)	9	4	44.4	16	9	56.3

^1^ Chronic obstructive pulmonary disease (COPD). ^2^ Familial hypercholesterolemia (FH). N, sample size in each group; n, absolute total number of elderly people; %, percentage of elderly people.

**Table 5 ijerph-18-08008-t005:** Meta-analysis of factors associated (quantitative variables) with mortality.

Variables	RR (95% CI)	I^2^	Z	*p*-Value
Male	0.98 (0.67; 1.43)	89.3	0.10	0.919
Chronic diseases	1.20 (0.94; 1.54)	-	1.48	0.139
Cancer	1.60 (0.60; 4.23)	-	0.92	0.356
Diabetes	1.90 (1.53; 2.37)	62.7	5.73	<0.001
Cardiovascular diseases/coronary artery disease	1.80 (0.85; 3.80)	92.0	1.53	0.125
COPD ^1^	2.19 (1.54; 3.10)	0.0	4.39	<0.001
Immunodeficiencies	5.28 (0.26; 108.12)	-	1.08	0.280
Chronic kidney disease	3.96 (2.65; 5.91)	0.0	6.73	<0.001
Metabolic disease	1.51 (0.60;3.75)	-	0.89	0.374
Obesity	1.28 (0.78; 2.10)	60.8	0.99	0.322
Hypertension	1.37 (1.24; 1.51)	69.3	6.25	<0.001
FH ^2^	3.27 (2.49; 4.29)	-	8.55	<0.001
Dementia	3.67 (2.43; 5.55)	-	6.17	<0.001
Smoking	0.74 (0.32;1.71)	-	0.70	0.483

^1^ Chronic obstructive pulmonary disease (COPD). ^2^ Familial hypercholesterolemia (FH). RR, relative risk; Z, Z statistic of meta-analysis; I^2^, I-square; 95% CI, 95% confidence interval.

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
