# Peer review of "Factors Associated with Mortality among Elderly People in the COVID-19 Pandemic (SARS-CoV-2): A Systematic Review and Meta-Analysis"

_ijerph, 2021, doi:10.3390/ijerph18158008_

Round 1

Reviewer 1 Report

Dear authors,

I agree with you reggarding the limitation of your meta-analysis since there are a small numbers of articles found in Italy. However, I have reviewed different meta-analyses only with three manuscripts found. It is a limitation but it is not weird.

I think this review, at this stage, can be usuful, as well as futrure ones reggarding this topic. The authors got a friendly-reading manuscript, brief sometimes, but enough. The introduction reflects clearly, at the end as it has to be, the goals of the manuscript.

Hoewer I have some minor questions related to Table 1 formating and, overall, related to the variables included in Table 2 and Table 3.
Regarding Table 1, please review the first column because you don't write the author nor the year.
Tables 2 an 3 only show information related to Age and Charlson Index, but you inform previously that the quantitative variables considered were more than these two ones (Lymphocytes, CRP, glucose, etc). Where is the information and meta-analyzes of these variables? or, why are not information related to them?
I think that providing information of meta-analyses of these variables would improve the quality of this systematic review, because only a metanalysis with Age and Charlton Index, I am no sure if it is enough.

Reviewer 2 Report

Proposed paper is interesting and well written. Data were correctly metaanalized and I have only two minor point that need to be addressed before it Can be accepted for pubblication:

  • Could it be possibile thatcomorbidities found to be related with mortality are The One evaluated in more study and also the most frequent One? In another word, it is possibile that something like a "sample size" issue affect the possibility to found other Association? Please discuss.
  • Older and frail patients presents a higher probability of presenting covid related cardiovascolare complication. Please comment in The discussione and cite also a very recently published review on this specific topic (High Blood Press Cardiovasc Prev. 2021 Jun 26:1–7.)

Author Response

Reviewer 2

“Proposed paper is interesting and well written. Data were correctly metaanalized and I have only two minor point that need to be addressed before it Can be accepted for pubblication:

Could it be possibile that comorbidities found to be related with mortality are The One evaluated in more study and also the most frequent One? In another word, it is possibile that something like a "sample size" issue affect the possibility to found other Association? Please discuss”.

Answer: Considering that these diseases are the most prevalent in the elderly population, it is expected that they have a greater relationship with the increase in mortality from COVID-19. This information has been added to the text as suggested.

“Older and frail patients presents a higher probability of presenting covid related cardiovascolar complication. Please comment in The discussion and cite also a very recently published review on this specific topic (High Blood Press Cardiovasc Prev. 2021 Jun 26:1–7.)”

Answer: In fact, cardiovascular diseases are among the greatest risk factors for increased mortality in covid-19, especially in the frail elderly. Therefore, we accept the suggestion and include the suggested reference.

This manuscript was submitted to a review in English by the MDPI's own editing service.

Round 2

Reviewer 1 Report

In my first review report I only asked for two minor considerations that have been taken into account by the authors. Then, this manuscript could be proposed for publication.

This manuscript is a resubmission of an earlier submission. The following is a list of the peer review reports and author responses from that submission.

Round 1

Reviewer 1 Report

The article reviews factors associated with COVID-19 related mortality. The research is not very original, since a similar topic has already been examined by other studies. Limitations of the study relate to the following:

  1. Only studies from Brazil and Italy were selected. Reason for this choice is not clear. In addition, the search did not identify any studies from Brazil and therefore the review is based only on studies published in Italy
  2. Risk factors were analysed in the population 65 or older. Reason for this choice is not clear. Numerous COVID-19 deaths were observed in younger population, which should be included in the review
  3. Several risk factors were examined by a single study and therefore it is difficult to draw conclusion on them
  4. I am surprised to see that age was not included among risk factors for mortality (table 5). How can this be explained?
  5. Frailty is an important risk factor in older people, but it was not assessed in the present study

Reviewer 2 Report

Dear authors, your manuscript is interesting, however, some comments should be taken into account to improve the work.

Regarding introducction, once you read the goals al lines 51-54 it is not adecuated, in my opinion, twenty more lines. If the authors consider, I would recommend to move the goals paragraph at the end of this section.

Perhaps a metaanalysis in a so short time period march-july 2020 is still a not accurate vision of the reality under analysis. A sentence justifying the relevance of this period in the introduccion as well in the limitations of the study could be of interest.

When the authors list the quantitative variables in Table 2 there are some weird number that should be pointed out. Variables CRD, CK, LDH, HscTnl, Ferritin among others have very high standar deviation values, por example CK has a SD of 195.17 and Ferrint has a SD of 1497.22. These are vey high values that could provide not real information, it may be because of the presence of outliers or measure errors. In may opinion the authors should investigate and justify the presence of these high values.

On the other hand when the authors introduce te qualitative variables in Table 4 it is not clear, at least for me, why the % of survivors and non-survivors of each record (row) does not sum 100%. Anyway there is a typographic mistake in the columns of Survivors because the authors use N twice. In this sence, there is another typographic mistake in line 205 because the authors write 'quantitative' when it shoud be written 'qualitative'.

Please review the citations in text because sometimes the authors don't put a space before the parentheses.

Finally, the discussion does not reflect the relevance of the results listed in the previous section, since the author don't mention any of the variables identified as risk factors.